# Dynamic Efficient Adversarial Training Guided by Gradient Magnitude

**Fu Wang**
Department of Computer Science
University of Exeter
Exeter, EX4 4QF
fw377@exeter.ac.uk

**Yanghao Zhang**
Department of Computer Science
University of Liverpool
Liverpool, L69 3BX
yanghao.zhang@liverpool.ac.uk

**Yanbin Zheng**
School of Mathematical Sciences
Qufu Normal University
Qufu, 10587
zheng@qfnu.edu.cn

**Wenjie Ruan**\*
Department of Computer Science
University of Exeter
Exeter, EX4 4QF
w.ruan@exeter.ac.uk

\* **Corresponding author**

## Abstract

Adversarial training is an effective but time-consuming way to train robust deep neural networks that can withstand strong adversarial attacks. As a response to its inefficiency, we propose Dynamic Efficient Adversarial Training (DEAT), which gradually increases the adversarial iteration during training. We demonstrate that the gradient's magnitude correlates with the curvature of the trained model's loss landscape, allowing it to reflect the effect of adversarial training. Therefore, based on the magnitude of the gradient, we propose a general acceleration strategy, M+ acceleration, which enables an automatic and highly effective method of adjusting the training procedure. M+ acceleration is computationally efficient and easy to implement. It is suited for DEAT and compatible with the majority of existing adversarial training techniques. Extensive experiments have been done on CIFAR-10 and ImageNet datasets with various training environments. The results show that the proposed M+ acceleration significantly improves the training efficiency of existing adversarial training methods while achieving similar robustness performance. This demonstrates that the strategy is highly adaptive and offers a valuable solution for automatic adversarial training.

## 1 Introduction

While deep neural networks (DNNs) continue to achieve better performance across a broad range of activities, their safety and dependability are getting increasing attention. These sophisticated models are known to be vulnerable to adversarial examples [6, 10, 16, 29], maliciously perturbed examples that can fool or mislead target DNNs but remain visually indistinguishable for humans, and adversarial training has been demonstrated as one of most effective defensive strategies to improve DNNs' adversarial robustness [2, 24]. The basic idea of adversarial training was proposed by Goodfellow et al. [12], which showed that the robustness of DNNs could be improved by injecting the adversarial examples into training data. Madry et al. [19] later formulated adversarial training as a Min-Max

36th Conference on Neural Information Processing Systems (NeurIPS 2022).

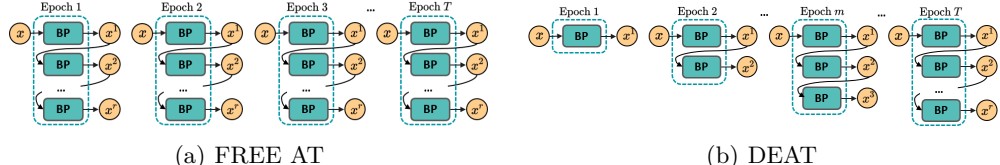

(a) FREE AT                                           (b) DEAT

Figure 1: An Illustration of the acceleration brought by Dynamic Efficient Adversarial Training (DEAT). While FREE Adversarial Training (AT) performs a fixed number of adversarial iterations, DEAT enables a more efficient training paradigm by executing less backpropagation (BP), which is the most computationally expensive operation in AT.

optimisation problem (see Eq. (1)), which indicates that using high-quality adversarial examples as training data can boost the robust performance of trained models. However, it usually takes multiple backpropagations to generate such adversarial perturbations, which is computationally expensive. The prohibitive training cost has become one of the major challenges in adversarial training [34].

Aiming to improve the training efficiency, in this paper, we take a closer look at existing adversarial training strategies [19, 26, 33] and find that redundant adversarial iterations could slow down the training process and sometimes even cause catastrophic forgetting [26], which spoils the trained model's robustness. To reduce the redundant backpropagation, we design a flexible adversarial training strategy, Dynamic Efficient Adversarial Training (DEAT), which begins the training with natural training data and gradually increases adversarial iterations when a pre-defined criterion is satisfied. Moreover, we show that the magnitude of a network's gradient with respect to training adversarial perturbations can roughly reflect the training effect and demonstrate that the model tends to gain 'robustness' after training if the gradient magnitude stays small. Based on such an observation, we design a self-adaptive criterion based on gradient magnitude for DEAT and further extend it as a general acceleration strategy that can directly boost the training efficiencies of other state-of-the-art adversarial training methods like [32, 38]. Experiments have been done on CIFAR-10 [17] and ImageNet [8] datasets with different model architectures, and the results demonstrate that our magnitude-guided strategy can significantly speed up adversarial training methods under different training environments and achieves comparable robustness performance.

## 2   Background and Related Works

**General Notations**   Given a dataset $D$ that contains $N$ pairs of example and label $(x_i, y_i)$, we denote by $X$ a set of all $x \in D$, where the number of examples in $X$ is written as $|X|$. Denoted by $F : \mathbb{R}^n \to \mathbb{R}^C$, a neural network classifier parameterized by $\theta$ takes $x \in \mathbb{R}^n$ as input and labels it into one of the $C$ classes based on $\arg\max_{c \in \{1,...,C\}} F_c(x; \theta)$. We denote by $L$ the loss function that measures the difference between the model's predictions and ground truth labels and write $f(x; \theta)$ as a shorthand of $L(F(x; \theta), y)$ for simplicity. This paper focus on the $\ell_\infty$ threat model [19], *i.e.*, $\delta \in B_\epsilon$ is an adversarial perturbation added on $x$ and $B_\epsilon$ is a $\ell_\infty$ norm ball with radio $\epsilon$. A projection is represented by $\mathcal{P}_\epsilon$, which maps any input $\delta$ into $B_\epsilon$.

**Adversarial Training**   Adversarial Training (AT) produces robust classifiers by optimising a Min-Max problem,

$$\min_\theta \mathbb{E}_{(x,y) \in D} \big( \max_{\delta \in B_\epsilon} f(x + \delta; \theta) \big). \tag{1}$$

Based on Projected Gradient Descent (PGD) adversarial attack [19], PGD AT first performs multi-step PGD attacks to approximately solve the inner maximisation problem in Eq. (1), where the perturbation crafted by PGD at $i$-th iteration can be described as

$$\delta^i = \mathcal{P}_\epsilon \Big( \delta^{i-1} + \alpha \cdot \text{sign} \big( \nabla_x f(x + \delta^{i-1}; \theta) \big) \Big). \tag{2}$$

The produced adversarial examples are utilised as training data, and the model parameter $\theta$ is then updated to minimise the training loss.

Table 1: A high-level comparison with previous effect adversarial training methods

| Methods | Acceleration strategy | Model independent | Adaptive training | Work with FAST training | Examined on ImageNet |
|---|---|---|---|---|---|
| FREE [26] | Update model parameters and adversarial perturbation simultaneously | ✓ | ✗ | ✓ | ✓ |
| YOPO [37] | Only attack the first layer of trained model to produce training adversarial examples | ✗ | ✗ | ✗ | ✗ |
| Amata [35] | Linearly increasing the number of adversarial iterations | ✓ | ✗ | ✓ | ✓ |
| U-FGSM [33] | Single adversarial step with large step size | ✓ | ✗ | ✓ | ✓ |
| FAT [39] | Conduct early-stop when generating training adversarial examples | ✓ | ✓ | ✗ | ✗ |
| This paper | A self-adaptive dynamic adversarial training framework | ✓ | ✓ | ✓ | ✓ |

Although PGD AT can withstand strong adversarial attacks, their computational cost is linear to the number of additional adversarial iterations [33], which is extremely computationally expensive in practice. Therefore improving its efficiency has become an active research topic [3], while reducing the number of iterations would be the most straightforward strategy to accelerate the training procedure. In Dynamic Adversarial Training (DAT) [31], the number of adversarial iterations is gradually increased according to a first-order stationary condition that measures the training effect. However, DAT needs to initialise the increasing threshold on a pre-trained model with a similar architecture. Based on an annealing mechanism, Amata linearly increases adversarial steps and adjusts the adversarial step size during training [35]. As we will show later, determining the optimal training schedules for different training tasks remains difficult. Zhang et al. [37] proposed YOPO, which simplifies the adversarial iteration by only generating adversarial examples toward the first layer of the trained model for adversarial training. Friendly Adversarial Training (FAT) carries out PGD adversaries with an early-stop mechanism to prevent overfitting and reduce training cost [39]. Wong et al. [33] found that one step PGD attack, also known as First Gradient Sign Method (FGSM) [12], with large step size and uniform initialisation, could achieve comparable performance to regular PGD AT and be significantly more efficient. They also reported that cyclic learning rate [28] and half-precision computation [20] could significantly boost training efficiency. We refer to training with cyclic learning rate and half-precision computation as FAST training [33] and to FGSM AT with uniform initialisation as U-FGSM in the rest of this paper. Instead of improving PGD AT, Shafahi et al. [26] proposed a novel training method called FREE, which makes the most use of backpropagation by updating the model parameters and adversarial perturbation simultaneously. This mechanism is called batch replay, and we visualise the process in Fig. 1(a). Table 1 summarises recent progress in efficient AT and shows the differences between our method and other approaches.

In parallel to efficiency, many efforts have been made to improve AT's effectiveness, *e.g.*, [1, 22, 24]. In this direction, some advanced surrogate loss functions have been proposed to enhance or replace the cross-entropy loss. TRADES can be viewed as a combination of the cross-entropy loss and a KL-divergence penalty [38]. On top of TRADES, MART merges the cross-entropy loss and the margin loss and pays more attention to misclassified examples [32]. Moreover, Zhang et al. [39] show that these advanced surrogate losses can also benefit from efficient AT methods.

**Lipschitz Condition**    The Lipschitz constant for $f(\cdot; \theta)$ gives an upper bound on how fast the loss value changes when small perturbations are added to the network's input [29]. This concept is closely related to the robustness of DNNs.

**Definition 1** (Lipschitz continuity). *Let $\| \cdot \|_a$ and $\| \cdot \|_b$ be two norms on $\mathbb{R}^n$ and $\mathbb{R}$, respectively. We say $f : \mathbb{R}^n \to \mathbb{R}$ is Lipschitz continuous if there exists a Lipschitz constant $K > 0$ such that*

$$\|f(x) - f(x + \delta)\|_b \leq K \|\delta\|_a, \ \forall \delta \in B_\epsilon.$$

**Assumption 1** (Sinha et al. [27]). *Let $\| \cdot \|_p$ be the dual norm of $\| \cdot \|_q$. $f(\cdot; \theta)$ satisfies the Lipschitzian smoothness condition*

$$\|\nabla_x f(x; \theta) - \nabla_x f(x + \delta; \theta)\|_p \leqslant l_{xx} \|\delta\|_q, \ \forall \delta \in B_\epsilon, \tag{3}$$

*where $l_{xx}$ is a positive constant.*

**Assumption 2** (Deng et al. [9]). *Given $f(\cdot; \theta)$ and an example $x$, there exists a stationary point $x + \delta^* \in B_\epsilon$ such that $\nabla_x f(x + \delta^*; \theta) = 0$.*

Given by Definition 1, in this paper, we suppose that the network $F$ and loss function $L$ are Lipschitz continuous [4, 25]. The Lipschitz condition Eq. (3) proposed by Sinha et al. [27] is commonly used by recent studies [31, 35]. Meanwhile, Deng et al. [9] proved that the gradient-based adversary could find a local maximum that supports the existence of $\delta^*$.

## 3 Gradient Magnitude Guided Adversarial Training

Previous studies implicitly show that AT is a dynamic procedure [5, 31], which means the number of adversarial iterations to generate suitable perturbation is floating. Thus, we wish to push the efficiency of AT by conducting as few adversarial iterations as possible.

### 3.1 Motivation and Naive Implementation

To provide an intuitive understanding of the dynamics of AT, we illustrate the learning process in which learned models become more robust. We perform three common AT strategies with ten training epochs and the cycle learning rate [28] and evaluate the robustness of trained models via a PGD attack with twenty iterations, denoted as PGD-20. The three methods are U-FGSM, PGD AT, and FREE, where U-FGSM is the one-step AT; PGD AT, represented by PGD-AT-7, consists of seven adversarial iterations; FREE is denoted by FREE-4 or FREE-8, indicating that it is performed with four or eight batch replays, respectively. During FREE AT, the network is trained on clean examples in the first batch replay, and the actual adversarial training starts at the second iteration. However, if the model is adversarially trained repeatedly too many times on a mini-batch, catastrophic forgetting would occur and influence training efficiency [26]. As shown in Fig. 2, we noticed that the robustness of the FREE-8 trained model swings dramatically throughout adversarial training, the number of iterations required to generate an adequate perturbation appears to fluctuate, and redundant adversarial iteration may also damage the trained models' robustness.

**Naive implementation**   Motivated by the previous observation, we then introduce the Dynamic Efficient Adversarial Training (DEAT) framework, where we wish to push the training efficiency by removing redundant adversarial iteration. Similar to FREE, DEAT starts the training on natural examples by setting the number of batch replays $r = 1$ and gradually increases the number of batch replays to conduct adversarial training in the following training epochs (Pseudocode is presented in Appendix A).

DEAT requires a pre-defined criterion to determine the timing of increasing batch replay, which can be achieved in different ways. Intuitively, performing one extra batch replay every $d$ epoch is a practical and straightforward approach to achieving dynamic adversarial training. We implement DEAT to increase the number of batch replays every 3 epochs, denoted by DEAT-3, and present the training process in the third column of Fig. 2. Because DEAT-3 does not perform adversarial training in the first few epochs, the trained models achieved high classification accuracy on clean examples but did not gain robustness. Despite the absence of adversarial robustness in the initial few epochs, the robustness performance of the DEAT-3 trained model shoots up after just one epoch of training on adversarial training examples. In addition to increasing batch replay every few epochs, we notice that training accuracy can also be used to decide when to adjust the number of adversarial iterations. One can also observe from Fig. 2 that as FREE-8's robustness performance diminishes over epoch 2 to epoch 7, its training accuracy rises to 60%, which is remarkably high compared to other techniques, including DEAT-3, whose training accuracy is only slightly above 43%. On the other hand, while the FREE-8-trained model gains robustness during epochs 7 to 10, its training accuracy declines substantially to about 47%. Inspired by this phenomenon, we design another naive implementation of DEAT based on the training accuracy, *i.e.*, adding one more batch replay when the model's training accuracy exceeds the pre-defined threshold. We set the threshold at 40% and execute DEAT accordingly, signified by DEAT-40%. As shown in the third column of Fig. 2, DEAT-40% achieves a considerable performance with a smoother learning procedure compared to DEAT-3.

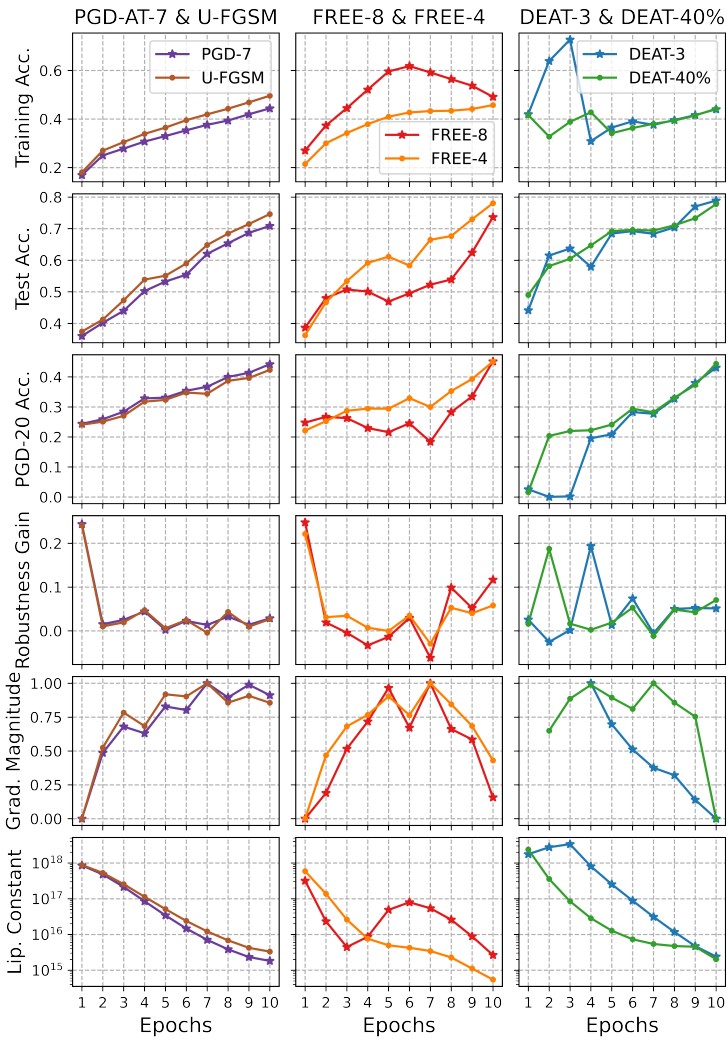

Figure 2: An illustration of the training process of six AT methods. The robustness is tested by the PGD-20 attack, and the fourth row illustrates how the trained models' robustness changes compared to the previous epoch. When the robustness gain in one epoch is larger than zero, the trained model is more robust than it was in the previous epoch. The gradient magnitude calculated via Eq. (5) is shown in the fifth row. The values are normalised for presentation, and a smaller gradient magnitude seems to indicate a more effective training epoch. We omit the gradient magnitude when the models are only trained on natural examples. The final row displays the upper bound of the Lipschitz constant, which is calculated via the product of each layer's Frobenius norm [4].

Table 2: The number of backpropagation conducted by adversarial training methods in ten epochs, where $M$ is the number of data batches in the training dataset.

| Methods | U-FGSM | PGD-AT-7 | FREE-4 | FREE-8 | DEAT-3 | DEAT-40% |
|---|---|---|---|---|---|---|
| #Backpropagation | $20M$ | $80M$ | $40M$ | $80M$ | $22M$ | $25M$ |

Alongside the robustness performance, it is natural to question DEAT's efficacy. Because the total number of backpropagation, the most computationally expensive operation in AT, can be viewed as a quantification of the training cost [33], we summarise the number of backpropagation of AT methods presented so far in Table 2. Letting $M$ be the number of data batches, U-FGSM and PGD-AT-7 conducted $20M$ and $80M$ times backpropagation in ten training epochs. FREE-8 spent the same amount of computational cost as PGD-AT-7, while

FREE-4 was half cheaper ($40M$) in the same period. By contrast, DEAT-3 only conducted $22M$ times backpropagation, given by $\lceil T(T+d)/2d \rceil$, where $T$ is the total number of epochs, and DEAT-40% performed $25M$ times. Such a performance indicates that DEAT has a significant advantage in training efficiency.

Nevertheless, although both naive implementations perform well in the preliminary test, they are not sufficiently generalisable to various datasets and model architectures. For instance, when doing DEAT-40% on a different training dataset, the training accuracy may never exceed the predefined threshold. If DEAT-3 is conducted with 50 epochs, the last training epoch will have 17 batch replays, which is unacceptable. In order to effectively configure DEAT with these naive criteria for varied training tasks, extensive parameter tuning may be required, hence limiting DEAT's usefulness. To further improve DEAT's generalisability, we need an adaptive strategy that does not rely on manual setup.

## 3.2 Gradient Magnitude Guided Approach

Recent studies [18, 26, 30] show that the adversarial robustness property is closely related to the geometry of the loss landscape, where AT enhance the trained models' robustness via flatting its loss landscape. In this sense, we propose utilising the model's loss curvature as an indicator of the training effect.

**Theorem 1.** *Given a perturbed example $x + \delta$, where $x \in \mathbb{R}^n$ and $\delta \in B_\epsilon$, if Assumptions 1 and 2 hold, the Lipschitz constant $l_{xx}$ of the first-order derivative of $f(x + \delta; \theta)$ towards training example $x$ satisfies*

$$\frac{\|\nabla_x f(x + \delta; \theta)\|_1}{2n\epsilon} \leqslant l_{xx}. \tag{4}$$

Recall that $l_{xx}$ in Eq. (3), a Lipschitz constant of the first-order derivative, implies the curvature around a particular example on the loss landscape by definition. Given by Theorem 1, we show that Eq. (4) gives a reasonable approximation of the Lipschitz constant $l_{xx}$ of $\nabla_x f(x + \delta; \theta)$ within a small $\ell_\infty$ norm ball [14] (proof is presented in Appendix B). In practice, once the training set $X$ and adversarial threat model are fixed, $n$ and $\epsilon$ in Eq. (4) are constants. The flatness around one example can then be measured by $\|\nabla_x f(x + \delta; \theta)\|_1$, and we take an accumulation of the flatness of examples in $X$, *i.e.*,

$$l_X = \sum_{x \in X} \|\nabla_x f(x + \delta; \theta)\|_1, \tag{5}$$

to quantify the overall curvature of a model.

To see whether the gradient magnitude can reflect the training effect, we visualise the gradient magnitude in the fifth row of Fig. 2. For comparison, robustness gain, the changes in the trained models' robustness relative to the previous epoch, and a loose Lipschitz constant of $f(\cdot; \theta)$ at the end of each training epoch are also visualised in the fourth and sixth rows of Fig. 2. Through an epoch-by-epoch comparison in Fig. 2, we can see that the gradient magnitude precisely reflects each adversarial training epoch's effectiveness. When the robustness of the models increases, the value of gradient magnitude is moderate, but when the robustness decreases, the gradient magnitude of that epoch increases dramatically. In contrast, although a more robust model tends to have a smaller first-order Lipschitz constant [29], the latter can only indicate roughly that an adversarially trained model is becoming more robust but is unable to identify which training epoch was more effective.

While both theoretical and empirical analysis suggests that the trained model gains robustness effectively when gradient magnitude $l_X$ maintains small, how to quantify a threshold for $l_X$ to enable self-adaptive AT remains a question. DAT [31] initialises its threshold via conducting a weak adversarial attack on pre-trained models, while the success of U-FGSM [33] demonstrates that one step attack with uniform initialisation could generate qualified training perturbation. Inspired by these previous works, we initialise the increasing threshold to $l_X$ at the second epoch that only contains one adversarial iteration. As described in Algorithm 1, we slightly enlarge the threshold via a relax parameter $\gamma > 1$ at initialisation and each update. By doing so, the number of adversarial iteration $k$ will only be increased when the current $l_X$ is strictly larger than in previous epochs. Because $\nabla_x f(x + \delta; \theta)$ has already been

**Algorithm 1** M+: gradient magnitude guided adversarial training.

---

**Require:** Training set $X$, total epochs $T$, adversarial radius $\epsilon$, step size $\alpha$, the number of mini-batches $M$, the number of adversarial iteration $k$, the evaluation of current training effect $l_X$ and a relax parameter $\gamma$

1:  $k \leftarrow 0$
2: **for** $t = 1$ to $T$ **do**
3:     $l_X \leftarrow 0$
4:     **for** $i = 1$ to $M$ **do**
5:         Craft adversarial perturbation $\delta_i$ via $k$ backpropagation.
6:         Save the gradient $\|\nabla_x f(x_i + \delta_i; \theta)\|_1$ at the least iteration.
7:         $l_X \leftarrow l_X + \|\nabla_x f(x_i + \delta_i; \theta)\|_1$
8:         Update model parameter $\theta$.
9:     **end for**
10:    **if** $t = 1$ **then**
11:       $k \leftarrow k + 1$
12:    **else if** $t = 2$ **then**
13:       Threshold $\leftarrow \gamma \cdot l_X$                  ▷ Initialise threshold
14:    **else if** $l_X >$ Threshold **then**
15:       Threshold $\leftarrow \gamma \cdot l_X$; $k \leftarrow k + 1$        ▷ Update threshold
16:    **end if**
17: **end for**

---

computed in generating adversarial training examples, such a criterion only takes a trivial cost to enable the fully automatic training procedure. Furthermore, through the different implementations of the training procedure in lines 5 to 7 of Algorithm 1, we enable the M+ criterion to collaborate with various existing AT methods.

## 4   Experiments

This section evaluates the proposed methods on the CIFAR-10 and ImageNet datasets. All experiments are built with Pytorch [23] on a workstation that has an Intel i9-9820X processor and 128GB memory. For CIFAR-10 experiments, we use a GeForce RTX 2080Ti, while the ImageNet experiments run on 3 RTX 2080Ti graphics cards.

### 4.1   A benchmark under a fair training setup

To compare with recently efficient AT methods, we first present a benchmark on the CIFAR-10 dataset under the same training setup, where all environmental variables are under control except AT methods themselves, to make a fair comparison. In addition, TRADES [38] and MART [32] are adopted as advanced surrogate loss functions that can be accelerated [39].

We make a fair examination of all baselines on training PreAct-ResNet 18 [13] and DenseNet 121 [15], where all methods are carried out with 50 epochs and a multi-step learning rate schedule. The detailed setup of training variables and baselines is summarised in Appendix C.1. The number of adversarial iterations performed for perturbing training examples has a critical impact on the performance of baseline methods. Therefore, we adopt only the optimal configuration from their original implementations. As for robustness evaluation, we use PGD-$i$ to represent a PGD adversary with $i$ iterations, while CW-$i$ adversary optimises the margin loss [6] via $i$ PGD steps. The adversarial step size is $2/255$, and the adversarial radio $\epsilon$ is $8/255$. The standard AutoAttack [7], denoted by AA, is also included in our evaluation. Each adversarially trained model is tested by a CW-20 adversary on a validation set during training, and we save the most robust checkpoints for the final evaluation to avoid the robust overfitting [33].

The benchmark is summarised in Table 3. The loss and robustness of trained models are illustrated in Appendix C.4. How M+ methods adjust the number of adversarial iterations is visualised in Appendix D. On the PreAct-ResNet 18, which is one of the most commonly used model architectures in adversarial training literature, we can see that TRADES achieves the best performance on the AA test, but it is also the most time-consuming. M+TRADES

Table 3: Evaluation on CIFAR-10 under the standard training setup. We write 'F+' to represent an AT method accelerated by FAT and 'M+' to represent methods accelerated by M+ criterion.

| Model Architecture | Method | Natureal | Adversarial | | | Time |
| | | | PGD-100 | CW-20 | AA | (min) |
| --- | --- | --- | --- | --- | --- | --- |
| PreAct ResNet 18 | FREE-4 | **85.10%** | 47.17% | 47.11% | 43.98 | **83.1** |
| | FREE-6 | 82.30% | 48.27% | 47.12% | 44.78 | 124.0 |
| | FREE-8 | 80.77% | 48.54% | 47.11% | 44.79 | 162.4 |
| | M+DEAT | 83.93% | **48.87%** | **48.12%** | **45.56** | 102.1 |
| | PGD-AT-7 | 82.33% | 49.48% | **48.64%** | 45.95 | 157.2 |
| | FAT-5 | 81.77% | 48.65% | 47.97% | 44.87 | 92.2 |
| | YOPO-5-3 | 82.09% | 43.02% | 43.23% | 40.69 | **77.8** |
| | Amata-2-10 | 79.63% | **50.28%** | 48.24% | **46.20** | 135.4 |
| | M+PGD | **83.43%** | 45.94% | 45.66% | 42.57 | 94.8 |
| | TRADES | 79.29% | 51.06% | 48.46% | **47.64** | 456.4 |
| | F+TRADES | **81.91%** | 51.12% | **48.72%** | 47.54 | **236.0** |
| | M+TRADES | 79.89% | **51.43%** | 48.61% | 47.52 | 317.7 |
| | MART | 78.04% | 54.88% | 48.42% | 46.67 | 148.7 |
| | F+MART | **84.05%** | 49.46% | 47.02% | 45.46 | 240.5 |
| | M+MART | 78.04% | **55.83%** | **49.19%** | **47.20** | **87.6** |
| DenseNet 121 | FREE-4 | **82.81%** | 44.83% | 44.41% | 41.06 | **122.4** |
| | FREE-6 | 80.68% | 46.71% | 45.90% | **43.49** | 184.5 |
| | FREE-8 | 79.04% | **46.67%** | **45.76%** | 43.22 | 244.4 |
| | M+DEAT | 82.18% | 45.16% | 45.22% | 42.53 | 152.6 |
| | PGD-AT-7 | 71.19% | 43.40% | 41.50% | 39.54 | 220.9 |
| | FAT-5 | **75.43%** | 41.90% | 39.61% | 38.57 | **168.5** |
| | YOPO-5-3 | - | - | - | - | - |
| | Amata-2-10 | 72.10% | 43.43% | 41.90% | 39.77 | 177.6 |
| | M+PGD | 73.06% | **44.28%** | **43.16%** | **40.99** | 192.3 |
| | TRADES | 70.47% | 42.50% | 39.30% | 38.67 | 689.1 |
| | F+TRADES | **74.33%** | 42.75% | 39.65% | 39.05 | 499.1 |
| | M+TRADES | 72.32% | **43.34%** | **39.77%** | **39.15** | **451.8** |
| | MART | 63.19% | 44.62% | 39.61% | 38.05 | 330.9 |
| | F+MART | **77.39%** | 43.35% | 40.09% | 39.01 | 434.6 |
| | M+MART | 68.33% | **47.46%** | **42.08%** | **40.26** | **132.7** |

is 26% faster than TRADES with a similar overall performance, while F+TRADES obtains comparable robustness in half of TRADES's training time. According to Fig. 4(a) and Fig. 6(a), M+TRADES's training efficiency is comparable to F+TRADES's at the early and middle stages, but the increased adversarial steps slow it down at the final training stage. MART achieves the highest robust performance on the PGD-100 test and is notably faster than TRADES, but there is no substantial improvement on the CW-20 and AA tests. Our M+MART is 40% faster and achieves better robust performance, whereas F+MART is even slower and less robust than the original MART. In the PGD-based group, PGD-AT-7 is a strong baseline under the standard training setup [24], and it outperforms most accelerated AT methods on robustness tests except Amata. As shown in Fig. 6(a), M+PGD only conducts at most 4 backpropagations to generate training adversarial examples, which limits its final robustness performance. M+DEAT is about 35% faster than PGD-AT-7 and achieves comparable robustness regarding the CW-20 and AA tests. FREE-6 is more efficient than FREE-8 and shows similar robustness. M+DEAT is about 17% faster than FREE-6 and has higher accuracy on all evaluation tasks. As demonstrated by Fig. 4(a) and Fig. 6(a), M+DEAT performs fewer batch replays than other FREE methods at the early training stage, resulting in a large reduction in time consumption, which is to be expected.

Because DenseNet 121 have received less attention in the literature on adversarial training, existing AT approaches that rely on manual setup cannot perform adequately on this model. Our M+ acceleration, on the other hand, overcomes the limitations imposed by

manual adjustment and enables its cooperative methods to attain better performance in this benchmark. AT methods combined with our M+ acceleration achieve significantly better results than their original implementation regarding computing cost and AA score. How M+ methods change the number of adversarial iterations on DenseNet 121 is displayed in Fig. 6(b). Due to the space limitation, the evaluation under FAST training setup is presented in Appendix C.2, while the empirical analysis of hyper-parameters and ablation studies of training tricks are presented in Appendix C.3.

## 4.2 Evaluation on ImageNet

In this section, we evaluate the M+DEAT on ImageNet. Because the classical AT method could take thousands of GPU hours to adversarially train an ImageNet classifier [34], we consider FREE-4 as the baseline method, which finishes training within a reasonable amount of time and achieves better robustness than PGD AT [26]. Although U-FGSM can also work on ImageNet, its training procedure includes several tricks, *i.e.*, three training phases with different levels of cropping and learning rate schedules. Due to too many manual factors, U-FGSM is not considered as a baseline here. Because Ye et al. [35] did not release the implementation of the Amata method. We only list their result on ImageNet for accuracy comparison. To make a fair comparison, we use the same training setting as FREE-4's original implementation[1].

As shown in Table 4, M+DEAT can work properly on ImageNet, and the acceleration effect is considerable. Compared to FREE-4, M+DEAT is about 34% faster and shows similar levels of robustness when $\gamma = 1.1$. Because a small $\gamma$ could introduce more adversarial iterations, M+DEAT with $\gamma = 1.01$ is only 16% faster than FREE-4, but it achieves slightly better performance on robustness tests.

Table 4: Results of ResNet 50 on ImageNet ($\epsilon = 4/255$).

| Method | Natural | PGD-10 | PGD-50 | Time (h.) |
|---|---|---|---|---|
| FREE-4 | 61.02% | 31.90% | 30.93% | 52.43 |
| Amata-2-4 | 59.7% | 31.8% | 31.2% | - |
| M+DEAT | | | | |
| $\gamma = 1.1$ | 62.04% | 31.40% | 30.18% | 34.34 |
| $\gamma = 1.01$ | 60.10% | 32.34% | 31.41% | 44.67 |

## 5 Conclusion and Future Work

In this paper, we demonstrate that the gradient magnitude can indicate the effect of adversarial training and design a self-adaptive acceleration strategy accordingly. The proposed acceleration method, M+ acceleration, can easily collaborate with state-of-the-art adversarial training approaches to improve their training efficiency. Comprehensive experiments have been done on CIFAR-10 and ImageNet datasets to evaluate the efficiency and effectiveness of the proposed method, where M+ acceleration performs exceptionally well with various training setups, demonstrating its generalisability and adaptability.

We believe that the ultimate goal of this direction would be an efficient adversarial training framework that can work under multiple threat models on various deep learning tasks. However, this work has mainly explored $\ell_\infty$ norm-based adversarial training, so one of the future works is to generalise the magnitude guided strategy into threat models under other metrics. Besides, recent works [11, 21] proposed to constrain the gradient magnitude via regularisation, which could also improve the trained models' robustness. Although regularising gradient magnitude cannot affect the number of adversarial iterations directly, we would like to adopt and examine such regularisation methods in DEAT in the next step. The potential collaboration between our strategy and other efficient adversarial training methods like [35, 37] will also be explored in our future works.

---

[1]Code is available at `https://github.com/mahyarnajibi/FreeAdversarialTraining`

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

## A    Pseudocode of DEAT

---

**Algorithm 2** Dynamic Efficient Adversarial Training

---

**Require:** Training set $X$, total epochs $T$, adversarial radius $\epsilon$, step size $\alpha$, the number of mini-batches $M$, and the number of batch replay $r$

1: Initialise the number of batch replay: $r \leftarrow 1$         $\triangleright$ Start training on clean examples
2: **for** $t = 1$ to $T$ **do**
3:      **for** $i = 1$ to $M$ **do**
4:          Initialise perturbation $\delta_i$
5:          **for** $j = 1$ to $r$ **do**         $\triangleright$ Adversarial training starts when $j = 2$
6:             $\nabla_x, \nabla_\theta \leftarrow \nabla f\left(x_i + \delta_i; \theta\right)$
7:             $\theta \leftarrow \theta - \nabla_\theta$            $\triangleright$ Apply gradient descent
8:             $\delta \leftarrow \mathcal{P}_\epsilon(\delta + \alpha \cdot \text{sign}\left(\nabla_x\right))$
9:          **end for**
10:      **end for**
11:      **if** meet increase criterion **then**
12:          $r \leftarrow r + 1$        $\triangleright$ Add one more batch replay since the following epoch
13:      **end if**
14: **end for**

---

## B    Proof of Theorem 1

Recall that

**Theorem 1.** *Given a perturbed example $x + \delta$, where $x \in \mathbb{R}^n$ and $\delta \in B_\epsilon$, if Assumptions 1 and 2 hold, the Lipschitz constant $l_{xx}$ of the first-order derivative of $f(x + \delta; \theta)$ towards training example $x$ satisfies*

$$\frac{\|\nabla_x f\left(x + \delta; \theta\right)\|_1}{2n\epsilon} \leqslant l_{xx}.$$

*Proof.* Substituting $x + \delta$ and the stationary point $x + \delta^*$ into Eq. (3) gives

$$\|\nabla_x f(x + \delta; \theta) - \nabla_x f\left(x + \delta^*; \theta\right)\|_1 \leqslant l_{xx} \|\delta - \delta^*\|_\infty. \tag{6}$$

Because $x + \delta^*$ is a stationary point and $\nabla_x f\left(x + \delta^*; \theta\right) = 0$, the left side of Eq. (6) can be written as $\|\nabla_x f\left(x + \delta; \theta\right)\|_1$. The right side of Eq. (6) can be simplified as

$$\begin{aligned} l_{xx} \|\delta - \delta^*\|_\infty &\leqslant l_{xx} \left(\|\delta\|_\infty + \|\delta^*\|_\infty\right) \\ &\leqslant l_{xx} \cdot (2n\epsilon). \end{aligned} \tag{7}$$

The first inequality holds because of the triangle inequality, and the second inequality holds because both $\|\delta\|_\infty$ and $\|\delta^*\|_\infty$ are upper bounded by $B_\epsilon$ that is given by $n\epsilon$. So Eq. (6) can be written as Eq. (4). This concludes the proof. $\qquad \square$

## C    Additional Experiments

### C.1    Experiment setup in Section 4.1

This benchmark includes two model architectures, *i.e.*, PreAct-ResNet 18 and DenseNet 121. All models are trained with 50 epochs, where we use the stochastic gradient descent optimiser and a multi-step learning rate schedule, where the learning rate is decayed at 25 and 40 epochs. The initial learning rate for PreAct-ResNet 18 is 0.05. However, DenseNet 121 fail to converge when trained with the same learning rate as PreAct-ResNet 18. Therefore, we set their initial learning rate at 0.01. All training hyper-parameters are reported in Table 5.

In Tab. 3, FAT [39] is carried out with 5 maximum training adversarial iterations. According to [38, 39], both original TRADES and FAT with TRADES surrogate loss, denoted by F+TRADES, use 10 adversarial steps. The same number of adversarial steps is also applied

Table 5: Detailed setup for the standard training benchmark in Sec. 4.1

| Setup | | Model architectures | |
| --- | --- | --- | --- |
| | | PreAct ResNet 18 | DenseNet 121 |
| Training | Initial learning rate | 0.05 | 0.01 |
| | Total epochs | 50 | |
| | Learning rate schedule | Multi-step learning rate, where the learning rate is decayed at 25 and 40 epochs | |
| | Batch size | 128 | |
| Optimisation | Optimiser | Stochastic Gradient Descent | |
| | Momentum | 0.9 | |
| | Weight decay | 0.0005 | |

in the original MART and FAT with MART surrogate loss (F+MART), which is the same setup used by [32]. As for the hyper-parameter $\tau$, which controls the early stop procedure in FAT, we conduct the original FAT, F+TRADES and F+MART at $\tau = 3$, which enables their highest robust performance [39]. Furthermore, YOPO-5-3 [37] performs 3 backpropagations to update training adversarial examples and repeats such a procedure 5 times. Amata-2-10 [35] linearly increases the number of adversarial iterations from 2 to 10 during training. FREE is performed with three different numbers of batch replays, where FREE-8 conducts 8 batch replays and would show better robustness, while FREE-4 only does 4 batch replays and should be relatively more efficient [26]. All gradient magnitude accelerated methods, namely M+DEAT, M+PGD, M+TRADES, and M+MART, are performed at $\gamma = 1.01$, which is the relax parameter. For the purpose of reproducibility, we release the code at `https://github.com/TrustAI/DEAT`.

## C.2 Utilising the FAST training setup

To demonstrate M+DEAT's adaptability, we compare it with two naive DEAT methods, FREE-4, FREE-8, and PGD-AT-7 under the FAST Training Setup on CIFAR-10 and report the evaluation results. Recall that the FAST training setup was proposed by [33], which utilises the cyclic learning rate [28] and half-precision computation [20] to speed up adversarial training. Here, the maximum of cyclic learning rate is carried out at 0.1. As a reminder, DEAT-3 increases the number of batch replays every three epochs, and DEAT-40% determines the timing of increase based on the training accuracy. All DEAT methods are evaluated at step size $\alpha = 10/255$. We set $\gamma = 1$ for PreAct-ResNet and $\gamma = 1.01$ for Wide-ResNet [36]. All methods are carried out with 10 adversarial training epochs. Because DEAT methods start training on natural examples, DEAT-3 has 12 training epochs in total, and DEAT-40% and M+DEAT run 11 epochs. We report the average performance of these methods on three random seeds.

The results on PreAct-ResNet 18 are shown in Table 6, and we can see that all DEAT methods perform properly under the FAST training setup. Among the three DEAT methods, models trained by M+DEAT show the highest robustness. Because of the increasing number of batch replays at the later training epochs, M+DEAT performs more backpropagations to finish training, so it spends slightly more training time. FREE-4 achieves a comparable robust performance to M+DEAT, but M+DEAT is 20% faster and has higher classification accuracy on the natural test set. The time consumption of FREE-8 almost triples those two DEAT methods, but it only achieves a similar amount of robustness to DEAT-3. PGD-AT-7 uses the same number of backpropagations as FREE-8 but performs worse.

The comparison on Wide-ResNet [36] model also provides a similar result. The evaluation results on Wide-ResNet 34 architecture are presented in Table 7. We can see that most methods perform better than before because Wide-ResNet 34 is more complex and has more parameters than PreAct-ResNet 18. Although FREE-8 achieves the highest overall robustness, its training cost is much more expensive than others. Compared to FREE-8, FREE-4 is half cheaper than FREE-8 and also performs considerably. M+DEAT is 22% faster than FREE-4 with comparable robustness and achieves 97% training effect of FREE-8

with 40% time consumption. Other DEAT methods are faster than M+DEAT because fewer backpropagations have been conducted. Models trained by them show reasonable robustness on all tests, but their performance is not as good as that of M+DEAT. Please note that the experiment on Wide-ResNet is meant to demonstrate the generalizability of M+ acceleration, not to challenge the robustness record.

Table 6: Evaluation on CIFAR-10 with PreAct-ResNet 18 under FAST training setup.

| Method | Natural | Adversarial | | | | Time |
| | | FGSM | PGD-100 | CW-20 | AA | (min) |
| --- | --- | --- | --- | --- | --- | --- |
| DEAT-3 | **80.09%** | 49.75% | 42.94% | 42.26% | 40.17 | 5.8 |
| DEAT-40% | 79.52% | 48.64% | 42.81% | 42.08% | 40.06 | **5.6** |
| M+DEAT | 79.88% | **49.84%** | 43.38% | **43.36%** | **41.18** | 6.0 |
| FREE-4 | 78.46% | 49.52% | 43.97% | 43.16% | 40.65 | 7.8 |
| FREE-8 | 74.56% | 48.60% | **44.11%** | 42.49% | 40.72 | 15.7 |
| PGD-AT-7 | 70.90% | 46.37% | 43.17% | 41.01% | 40.29 | 15.1 |

Table 7: Evaluation on CIFAR-10 with Wide-ResNet 34 under FAST training setup.

| Method | Natural | Adversarial | | | | Time |
| | | FGSM | PGD-100 | CW-20 | AA | (min) |
| --- | --- | --- | --- | --- | --- | --- |
| DEAT-3 | 81.58% | 50.53% | 44.39% | 44.43% | 43.50 | 41.6 |
| DEAT-40% | 81.49% | 51.08% | 45.03% | 44.49% | 42.50 | **38.9** |
| M+DEAT | **81.59%** | 51.29% | 45.27% | 45.09% | 44.45 | 43.1 |
| FREE-4 | 80.34% | **51.72%** | 46.17% | 45.62% | 43.30 | 55.4 |
| FREE-8 | 77.69% | 51.45% | **46.99%** | **46.11%** | **44.90** | 110.8 |
| PGD-AT-7 | 72.12% | 47.39% | 44.80% | 42.18% | 41.95 | 109.1 |

**Comparing to U-FGSM** Here, we report the competition between DEAT and U-FGSM. The comparison is summarised in Table 8. We can see that the differences in robustness performance between U-FGSM and DEAT methods are subtle under the FAST training setup. They achieve a similar balance between robustness and time consumption, while M+DEAT performs better on most tests but also takes slightly more training time. When using the standard training setup, U-FGSM is faster but only achieves a fair robust performance.

## C.3 Empirical analysis

In this section, we explore the impact of two hyper-parameters, *i.e.*, step size $\alpha$ and relax parameter $\gamma$, and conduct ablation studies on several tricks involved in the benchmark under the standard training setup.

**The impact of hyper-parameters $\alpha$ and $\gamma$** Under the FAST training setup, we evaluate the trained models' performance with different step sizes at $\gamma \in \{1.1, 1.01, 1.001\}$. Each model is trained with 10 epochs, and each combination of $\alpha$ and $\gamma$ has been tested on 5 random seeds. To highlight the relationship between models' robustness and the corresponding training cost, which is denoted by the number of backpropagations (#BP), we use PGD-20 error that is computed as one minus the model's classification accuracy on PGD-20 adversarial examples. The result has been summarised in Fig. 3. This experiment shows that the models' robustness is more sensitive to $\alpha$ rather than $\gamma$. We can also see that M+DEAT achieves a slightly better robustness performance when using a large step size, e.g., 14/255, but the corresponding training cost is much higher than that with a lower step size when $\gamma$ is small.

**Ablation studies of three training tricks** The default training tricks in Table 3 are multi-step learning rate, uniformly Initialised perturbation, and slightly enlarge the step size $\alpha$. Under the standard training setup, we investigate whether these tricks are truly

Table 8: Compare M+DEAT to U-FGSM under different setups on CIFAR-10.

| Set up | Method | Natural | PGD-100 | CW-20 | AA | Time (min) |
|---|---|---|---|---|---|---|
| PreAct-Res. | M+DEAT | 79.88% | 43.38% | 43.36% | 41.18 | 6.0 |
| + FAST | U-FGSM | 78.45% | 43.86% | 43.04% | 39.80 | 5.6 |
| Wide-Res. | M+DEAT | 81.59% | 45.27% | 45.09% | 44.45 | 43.1 |
| + FAST | U-FGSM | 80.89% | 45.45% | 44.91% | 42.45 | 41.3 |
| PreAct-Res. | M+DEAT | 83.93% | 48.87% | 48.12% | 45.56 | 102.1 |
| + Standard | U-FGSM | 81.99% | 45.77% | 45.74% | 42.25 | 40.3 |

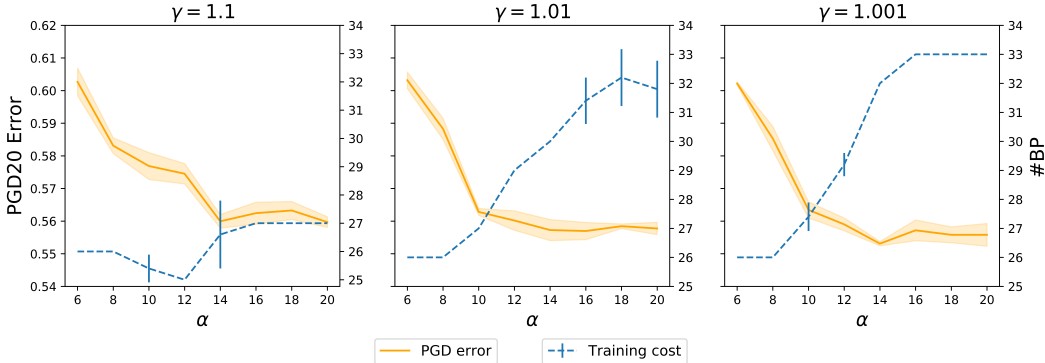

Figure 3: Visualise the impact of $\alpha$ and $\gamma$. #BP is a shorthand for the number of back-propagations. PGD-20 error is computed as one minus the model's classification accuracy of PGD-20 adversarial examples.

effective for M+DEAT. In addition to simply removing these tactics, we also evaluate an alternative for each. For the learning rate schedule, we consider the flat learning rate and cyclic learning rate. The perturbation initialisation is performed with zero initialisation and normally random initialisation. The step size $\alpha$ is additionally carried out at 8/255 and 12/255. It can be seen from Table 9 that the flat learning rate significantly weakens performance across the board, indicating that a dynamic learning rate schedule is necessary to accelerate training convergence. Comparing cyclic learning rate with multi-step learning rate, M+DEAT with the former achieves marginally better robustness performance with 10% longer runtime. In terms of robustness performance, uniform initialisation surpasses normal and zero initialisation, whereas zero initialisation is actually better than normal initialisation. The most appropriate step size is 10/255. In the meantime, enlarging the step size raises the computational cost of M+DEAT. Such an effect can also be observed in Fig. 3.

Table 9: Ablation studies of learning rate schedule, perturbation initialization, and step size with M+DEAT trained PreAct ResNet 18 on CIFAR-10.

| Learning rate schedule | | | Perturbation init. | | | Step size $\alpha$ ($\cdot$/255) | | | Natural | PGD-100 | CW-20 | Time |
|---|---|---|---|---|---|---|---|---|---|---|---|---|
| Multi-step | Cyclic | Flat | Uniform | Normal | Zero | 8 | 10 | 12 | | | | |
| ✓ | | | ✓ | | | ✓ | | | 83.72% | 47.97% | 47.66% | 92.8 |
| ✓ | | | ✓ | | | | | ✓ | 83.67% | 48.59% | 47.98% | 108.9 |
| ✓ | | | | | ✓ | | ✓ | | 81.95% | 49.27% | 47.65% | 105.1 |
| ✓ | | | | ✓ | | | ✓ | | **85.73%** | 45.87% | 46.37% | **81.6** |
| | ✓ | | ✓ | | | | ✓ | | 83.97% | **49.24%** | **48.65%** | 116.4 |
| | | ✓ | ✓ | | | | ✓ | | 66.20% | 35.41% | 34.95% | 100.3 |
| ✓ | | | ✓ | | | | ✓ | | 83.93% | 48.87% | 48.12% | 102.1 |

## C.4 Visualisation of training process in Sec 4.1

The adversarial training on PreAct-ResNet 18 is illustrated in Fig. 4 in terms of validation accuracy and training loss.

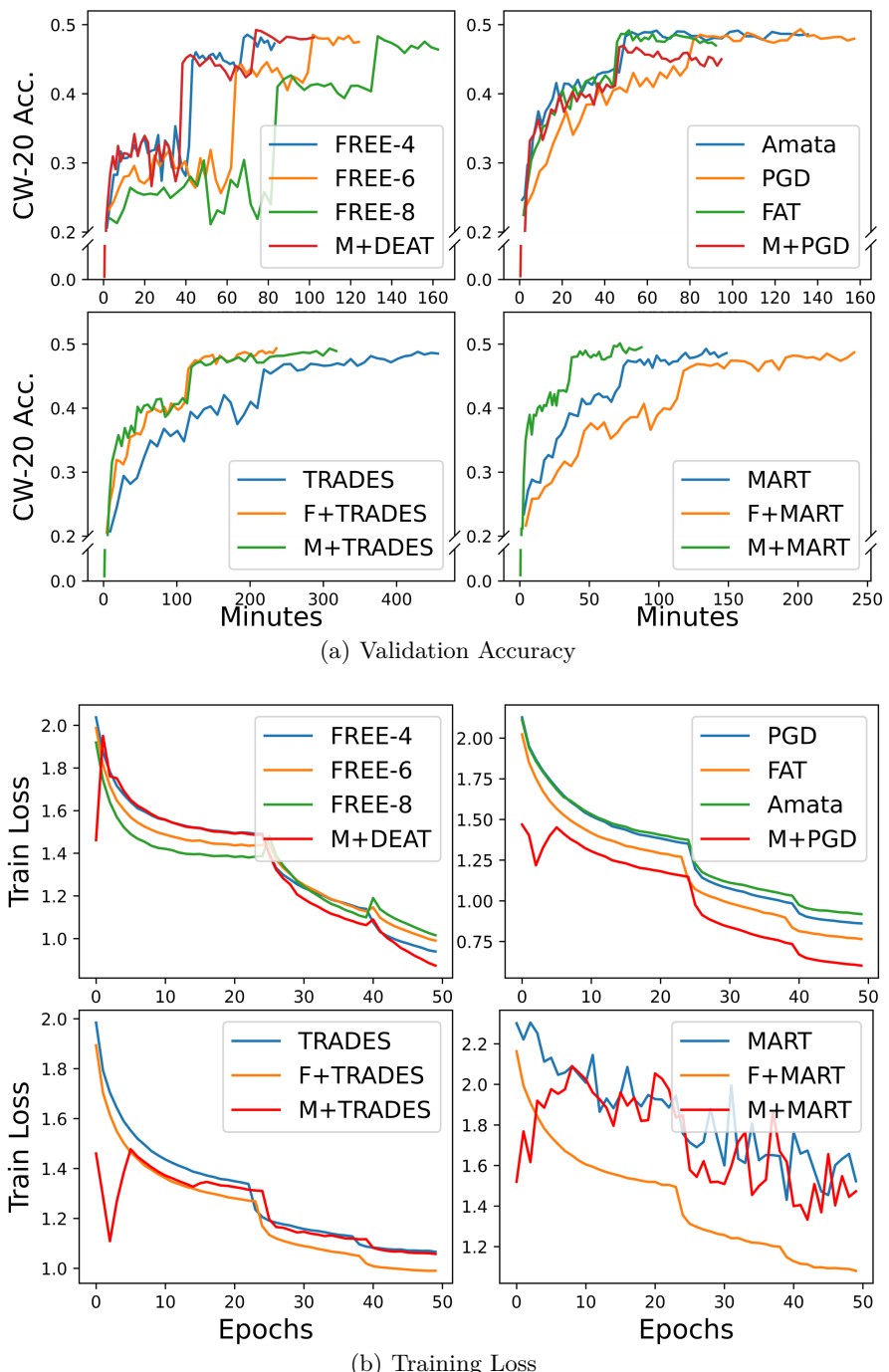

(a) Validation Accuracy

(b) Training Loss

Figure 4: An illustration of how adversarially trained PreAct-ResNet 18 gain robustness (a) and how their training loss changes (b).

The adversarial training on DenseNet 121 is illustrated in Fig. 4 in terms of validation accuracy and training loss.

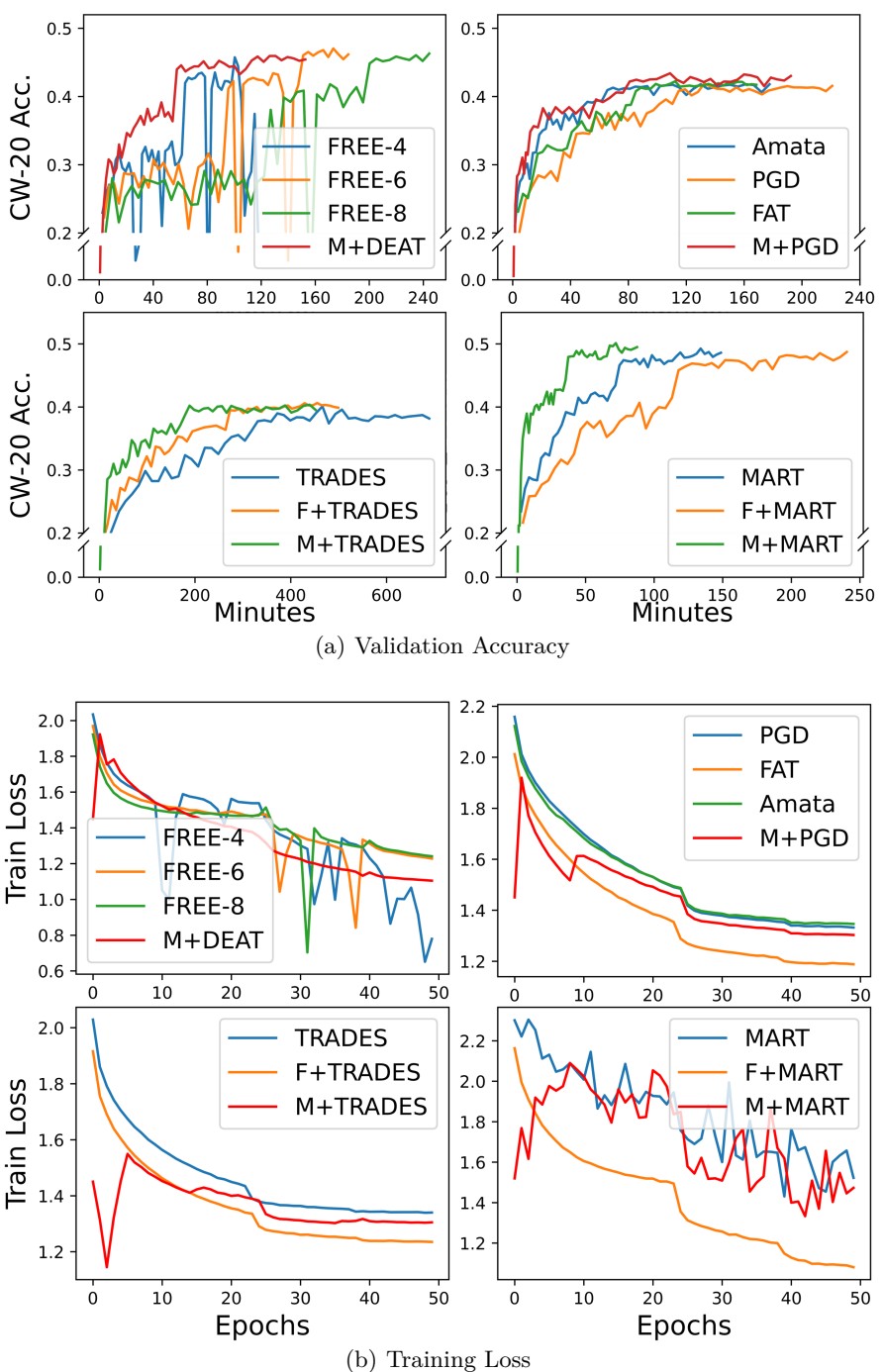

(a) Validation Accuracy

(b) Training Loss

Figure 5: An illustration of how adversarially trained DenseNet 121 gain robustness (a) and how their training loss changes (b).

# D  Visualisation of how the number of backpropagations change in M+ methods

M+ acceleration allows its cooperative methods to increase the number of adversarial iterations during training, resulting in less backpropagation and increased efficiency. To illustrate the adversarial training process of M+ techniques, we display in Fig. 6 the number of backpropagation operations performed by these methods.

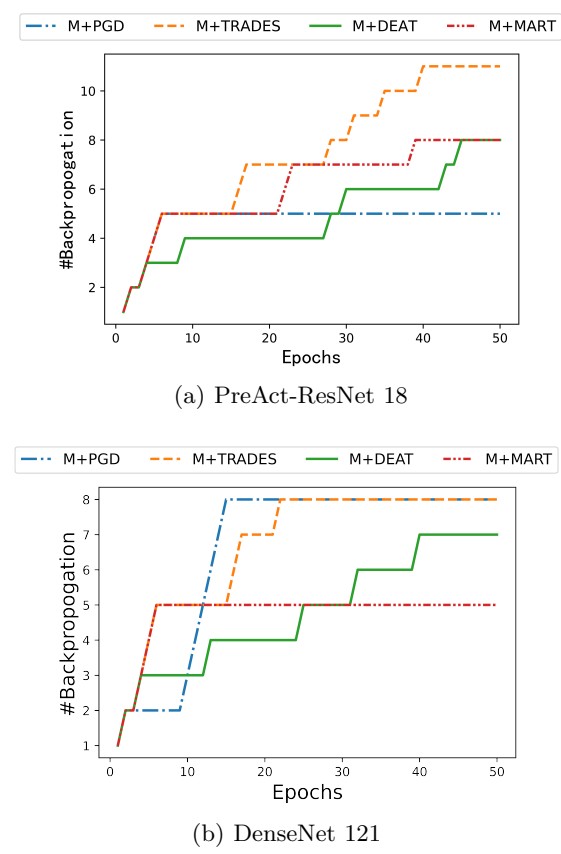

(a) PreAct-ResNet 18

(b) DenseNet 121

Figure 6:   The number of backpropagations with respect to epochs.

