# OpenReview forum: "Dynamic Efficient Adversarial Training Guided by Gradient Magnitude"
_NeurIPS.cc/2022/Workshop/TEA — TEA_

### Official Review · Reviewer_cUaz · 2022-10-16
**An efficient but unstable adversarial training method.**

**Rating:** 5
**Confidence:** 4

**Review:**

[Summary]

This paper proposes a method to make the adversarial training more efficient. The authors first propose a dynamic framework to adjust the adversarial iteration to speed up the training. Then, they further extend this method by adaptively tuning the adversarial iteration according to the gradient magnitude. Experimental results show that the proposed method could boost the adversarial robustness in some cases.

[Strengths]

This paper is well-organized and easy to follow. The authors start by introducing their motivation and a naïve implementation. Then they further improve the naïve implementation under the inspiration of some empirical findings.

[Weakness]

- It is unclear whether the comparison between different training methods in Section 3.1 is fair. Because the attacking settings in different training methods are different, the comparison between these methods may be unfair. For example, if the step size used in a training method is much larger than the step size used in another one, then the former training method may be very unstable, and the performance would be bad. Therefore, it would be better if the authors provided more details and discussed how to fairly compare these methods.
- The dynamic strategy in DEAT needs more explanations about why we need to increase the adversarial iteration. It is claimed that “if the model is adversarially trained repeatedly too many times on a mini-batch, catastrophic forgetting would occur and influence training efficiency” in Section 3.1. I think it suggests that increasing the adversarial iteration may raise the risk of catastrophic forgetting and hurt the efficiency. Then, it is confusing why the proposed DEAT increases the adversarial iteration when the training accuracy exceeds a certain threshold. The authors do not prove that increasing the adversarial iteration would boost the adversarial robustness.
- Figure 2 shows that the learning of DEAT is very unstable.
- The adversarial iteration in DEAT and the improved version guided by gradient magnitude keep increasing during the learning and do not have a limited scope. The experiments in the paper all use a small number of epochs. However, given a difficult learning task where the epoch number is large, the ultimate adversarial iteration may be very large, which leads to a high computational cost. It would be better if the authors discussed this flaw.
- It would be much better if the author could directly prove the correlation between the gradient magnitude and the robustness gain in training. The theoretical analysis in Section 3.2 only discusses the relationship between the gradient magnitude and its Lipschitz constant, and it is a very loose bound.
- In Table 4, the performance of the proposed method is very sensitive to the setting of $\gamma$. When $\gamma=1.1$, the performance of the proposed method is even worse than all baseline methods.

---

### Official Review · Reviewer_YGTt · 2022-10-16
**Gradient Magnitude Guided Adversarial Training**

**Rating:** 7
**Confidence:** 4

**Review:**

The authors propose an efficient and effective adversarial training method by guiding the training process with gradient magnitude. The authors provided theoretical and empirical analysis of why the proposed methods are more efficient. In general, the reviewer is fond of the idea and the presentation, but some issues must be improved further.

- It is difficult for the reviewer to extract meaningful information from Figure 2. Comparison between different methods is difficult due to the presentation and the analysis of the results is not detailed.
- The evaluation is comprehensive over different methods but the detailed parameters are not justified. For example, the total epochs and parameters are usually needed to be searched for different methods. Some latest work such as FreeLB is omitted. The reviewer suggests the authors add more related works and compare the performance with them as well.
- From table 3, it is clear that the proposed methods are not superior to all other methods. The reviewer would recommend the authors to tone down the claim in the introduction and abstract.

---

### Official Review · Reviewer_VfNq · 2022-10-19
**Reviewer's Comments**

**Rating:** 6
**Confidence:** 3

**Review:**

The paper proposed a new acceleration technique, termed M+ acceleration, for adversarial training (AT). As shown in CIFAR-10 and ImageNet evaluations, M+ acceleration could accelerate different variants of AT to a moderate level. The paper studied a vital problem and contained extensive experiment results. Besides these strengths, my additional comments are as below.

1) Speeding up adversarial training has raised increasing attention. Besides the conventional AT acceleration techniques reviewed in this paper, please try to cover more recent advancements. E.g., [ICML'22] "Revisiting and Advancing Fast Adversarial Training Through the Lens of Bi-Level Optimization."

2) The input gradient penalization was used in AT's variants, e.g., [https://arxiv.org/pdf/1905.11468.pdf] SCALEABLE INPUT GRADIENT REGULARIZATION FOR ADVERSARIAL ROBUSTNESS. Please clarify the differences between the proposed work and the existing one.

3) Strictly speaking, the curvature (or flatness of loss landscape) cannot be precisely characterized by a first-order gradient, e.g., [https://arxiv.org/pdf/1811.09716.pdf] Robustness via curvature regularization, and vice versa. Please add a more detailed discussion on Gradient Magnitude Guided Approach vs. curvature.

---

### Decision · Program_Chairs · 2022-10-21

**Decision:**

Accept

**Comment:**

The acceleration of adversarial training is an important research problem. The paper proposed a sensible solution supported by extensive experimental results. The reviewers find the paper well-written and easy to follow. The reviewers, especially Reviewer cUaz, pointed out several weakness regarding the experiments. Please try to address these issues in the final version if possible. Above all, the idea of the paper is still interesting in general, and will be of interest to the target audience of the workshop.